# On the prediction of underwater aerodynamic noise of offshore wind turbines

Laura Botero-Bolívar<sup>1</sup>, Oscar A. Marino Sánchez<sup>1</sup>, Martín de Frutos<sup>1</sup>, and Esteban Ferrer<sup>1</sup> <sup>1</sup>ETSIAE-UPM - School of Aeronautics, Universidad Politécnica de Madrid, Plaza Cardenal Cisneros 3, Madrid **Correspondence:** Laura Botero-Bolívar (l.botero@upm.es)

Abstract. The growing demand for offshore wind energy has led to a significant increase in wind turbine size and to the development of large-scale wind farms, often comprising 100 to 150 turbines. However, the environmental impact of underwater noise emissions remains largely unaddressed. This paper quantifies, for the first time, the underwater aerodynamic noise footprint of three large offshore turbines (5 MW, 10 MW, and 22 MW) and wind farms composed of these turbines. We propose a novel methodology that integrates validated wind turbine noise prediction techniques with plane wave propagation theory in different media, enabling turbine designers to predict and mitigate underwater noise emissions. Our results confirm that aerodynamic noise from offshore wind farms presents a potential environmental challenge, with negative effects on marine life. Addressing this issue is crucial to ensuring the sustainable expansion of offshore wind energy.

#### **1** Introduction

- 10 Anthropogenic underwater noise significantly impacts marine ecosystems, posing the natural equilibrium among different species Götz et al. (2009). Marine mammals are especially vulnerable to noise since they rely on sound to communicate, navigate, reproduce, feed, among other sensory purposes. A commonly cited problem concerning underwater noise is the masking problem (Erbe et al., 2016), where anthropogenic noise can mask sound naturally present in marine environments, leading to alteration of behavior, reduction of communication ranges, foraging, predator and habitat avoidance (Stöber and
- Thomsen, 2021; Daly and Harrison, 2012). Furthermore, the propagation of sound in the sea is enhanced by the higher speed 15 of sound in water than in air (1480 m/s for water vs. 343 m/s for air) and the lower sound attenuation in water (0.1 dB/km in seawater vs. 5 dB/km for air at 1 kHz). These physical factors, together with the channeling of sound in shallow waters, make anthropogenic noise more critical underwater. For example, Tougaard et al. (2020) measured underwater noise 20 km away from a small offshore wind farm composed of only 16 wind turbines. Therefore, predicting underwater noise produced by offshore energy devices is paramount to guarantee sustainable exploitation of energy sources.

5

Wind turbine design for offshore environments has primarily focused on maximizing energy production (e.g., Akhtar et al. (2024); Sherman et al. (2020); Desalegn et al. (2023)), with limited attention given to the acoustic footprint and its environmental impact. Specifically, the underwater aerodynamic noise generated by offshore wind turbines has not yet been quantified, which is the focus of this work.

<sup>20</sup> 

- Aerodynamic noise of wind turbines is caused by the interaction between the turbulent wind and the turbine blades. Thus, it depends on the operational conditions and size of the wind turbines the overall sound pressure level of a wind turbine scales with the fifth power with the wind turbine rotor diameter. In recent years, the size of offshore wind turbines has increased steadily in response to the growing demand for clean energy production. For example, the offshore wind turbine proposed within the IEA Task 55—REFWIND would produce 22 MW power with a rotor diameter of 284 m (Zahle et al., 2024) and the 30 recently announced MySE 22 MW offshore turbine from Mingyang Smart Energy with a rotor of 310+ meters (Mingyang-
- Wind-Power, 2024). In addition, we face a rapid increase in the number of turbines gathered in farms. Today, the largest offshore wind farms (e.g., London Array, Gemini, Hornsea Project One and Two (Ørsted, 2024)) include more than 150 turbines. Simple acoustics shows that the noise produced by a wind farm of N wind turbines scales with the factor  $20 \log_{10}(N)$ . The combination of increasingly large turbines gathered in farms with hundreds of turbines, combined with the negative impact of underwater
- noise, motivates this research: Can aerodynamic noise from offshore wind turbines affect marine life? To quantify this issue, we develop a new approach which combines wind turbine noise prediction techniques with wave theory to calculate the effective noise that penetrates underwater (due to the change of media).

Wind turbine noise may be classified as mechanical and aerodynamic acoustic noise. The first type has a defined tonal character and is produced by mechanical components such as the gearbox and bearings (and/or generator or cooling systems)

- located within the device nacelle and may be controlled/minimized by appropriate insulation of the nacelle. The second type is more complex and is caused by the interaction of the blades moving through the air. Previous studies on offshore wind turbine noise have only considered mechanical noise as it is directly propagated into the water through the vibrating tower (or platform) (Tougaard et al., 2020; Hooper et al., 2003; Madsen et al., 2006; Marini et al., 2017; Thomsen and Kafemann, 2006; Tougaard et al., 2009).
- A common justification for ignoring the aerodynamic noise (produced by the rotating blades) of offshore wind turbines is based on Snell's law (Chapman and Ward, 1990), which states that only one portion of the noise produced in the air propagates into the water. For air-water interfaces, Snell states that only sound waves within a cone of 13° angle with respect to the air-water interface normal vector can propagate into the water, see figure 2. This fact considerably limits the propagation of airborne noise sources into the water. Furthermore, the higher acoustic impedance of water compared to air (i.e., the acoustic
- impedance of water is 3600 times higher than the air's) leads to a high attenuation of the sound waves when entering water. The main contribution of this work is to show that despite this high attenuation, underwater noise is still potentially dangerous to marine animals. We couple wind turbine aerodynamic noise prediction methods with plane wave theory and Snell's law to predict wind turbine noise underwater. This contribution is a breakthrough for manufacturers and environmentalists, as it allows to consider noise generation in turbine design. Furthermore, The method is a fast turn-around method that can be
- used in low-fidelity models, but it can be easily combined with high-fidelity models to characterize the noise source or noise propagation.

Using the proposed approach, we will quantify how the underwater footprint of large offshore turbines can affect several marine species. We compute the aerodynamic noise of 5 MW, 10 MW, and 22 MW offshore wind turbines and compare

the underwater transmitted sound to the hearing thresholds of many marine animals. We confirm that these emissions are an environmental problem that is exacerbated when large offshore farms with hundreds of turbines are built.

The remainder of the paper is organized as follows. Section 2 presents the methodology used in the research, including wind turbine noise prediction, air-water interface modeling, and the characteristics of wind turbine models. Section 3 addresses the results and discussion. Finally, Section 4 shows the main conclusions of this research.

## 2 Methodology

#### 65 2.1 Wind turbine noise predictions

Aerodynamic noise, caused by the interaction of the flow with the structure, is the main source of noise of modern wind turbines Oerlemans et al. (2007). Figure 1 sketches the typical aerodynamic environment and the sources of noise of an offshore wind turbine. The atmospheric turbulence interacts with the leading edge of the rotating blades, causing a low-frequency noise, known as leading edge (LE) noise. Additionally, the turbulent boundary layer on the blades that interacts with the finite trailing

- edge causes mid- to high-frequency noise, referred to as trailing edge (TE) noise. Overall, the wide range of turbulent scales from hundreds of meters in atmospheric flow to millimeters in boundary layer flow encountered by the wind turbine blades cause aerodynamic noise to exhibit a broadband nature, covering a wide range of frequencies. This is particularly critical for marine environments, as it can affect a variety of marine species with different hearing thresholds.
- To compute the total aerodynamic noise of the wind turbine, we follow the method proposed by Schlinker-Amiet Schlinker 75 and Amiet (1981) for rotatory noise sources. We consider the strip theory approach, where the blade is divided into *n* segments. Each segment is considered as a 2D airfoil (as shown in the A-A cut in Figure 1). For each segment, leading- and trailing-edge noise ( $S_{pp|LE}$  and  $S_{pp|TE}$ ) are calculated as uncorrelated noise sources, such as:  $S_{pp|seg} = S_{pp|LE} + S_{pp|TE}$ .

Leading- and trailing-edge noise (LE and TE) are predicted using Amiet's theory (Amiet, 1975, 1976) and the extension of Roger and Moreau (2005) to consider the back-scattering effect caused by airfoils of finite chords. The blade is divided into

- segments that are more refined close to the blade tip, which is the part that generates most of the noise. An initial sinusoidal distribution of the location of the segments is proposed. The sinusoidal distribution is obtained by the horizontal coordinate of a point located in a semicircle of a diameter equal to the rotor radius (neglecting the inner part of the blades that consist of cylinders). The angle between the points in the radial axis was constant. After an iterative process to ensure that the aspect ratio (AR), defined as the span-chord ratio of the blade section, is larger than three, the radial position and the span- and
- chord-length distribution are obtained. The  $AR \ge 3$  condition is adopted to satisfy the far-field condition assumed in Amiet's theory.

The von Kármán model (von Kármán, 1948) calculates the inflow turbulence spectrum used as input for predicting LE noise and an extension of TNO-Blake model (Stalnov et al., 2016) computes the wall pressure spectrum to calculate trailing-edge noise. The boundary layer characteristics used as input in the TNO-Blake model are computed by XFOIL (Drela, 1989) using

the flow conditions (angle of attack,  $\alpha$  and relative velocity,  $U_{rel}$ ) obtained with the blade element momentum theory (BEMT). The transition for XFOIL simulations was fixed at 5% of the chord. BEMT solutions are obtained using the open-source code