# Peer review of "On the prediction of underwater aerodynamic noise of offshore wind turbines"

_Wind Energy Science, 2025_

## Author Comment (AC1)

We appreciate the kind comments and interesting points raised by the reviewer. We have answered their comments below.

1. Results should be summarized in the abstract:

We agree. We are now more specific in the abstract regarding the results found in the paper.

2. Line 27, it is not precise to say increasing with the rotor diameter, but better with the tip speed.

We agree with the reviewer that increasing the tip speed increases the wind turbine noise, and usually larger wind turbines have higher tip speeds. However, the tip speed is also related to the operational conditions, since a defined wind turbine could increase the tip speed (increasing the rotational speed). Additionally, increasing the wind turbine diameter increases the noise (even with the same tip speed) because of the larger span of the surface that is generating noise. Therefore, we do not believe that it is not precise to say that wind turbine noise increases with rotor diameter.

We think that we could make this statement a bit clearer, and we changed the statement in the text, specifying the contribution of the tip speed and the rotor diameter.

3. Line 43, not only mechanical noise, also the structure borne of aerodynamic noise.

The reviewer is totally right. We changed the sentence.

4. Line 91, "the transition was fixed at 5% of the chord", why didn't you use free-transition?

It is an excellent question. Due to the high Reynolds number of wind turbines, the boundary layer transition occurs near the leading edge. However, XFOIL is not very precise in calculating transition. Therefore, we believe that the most correct approach to calculate the boundary layer parameters for the wind turbine with XFOIL is setting a forced transition close to the leading edge, which is close to reality.

We have also checked the influence of the location of the forced transition (up to 20% of the chord), and the far-field noise of the turbines matches within 1 dB.

5. Figure 1, caption: Aerodynamic noise includes more than leading-edge and trailing-edge noise.

Yes. You are right, we meant the noise sources we considered predicting the wind turbine noise, which are also the most important. We have updated the label of the figure.

6. Line 95: check the Doppler effect factor for Spp.

We believe the reviewer means the factor  $(w_e/w)^2$ . However, we do not exactly understand what the author means by check.

If the reviewer means the exponent 2, there is a discussion in the literature regarding the exponent of the Doppler effect, whether it should be 1 or 2 (see Y. Rozenberg, M. Roger, S. Moreau, Rotating blade trailing-edge noise: Experimental validation of analytical model, AIAA journal 48 (5) (2010) 951–962 and S. Sinayoko, M. Kingan, A. Agarwal, Trailing edge noise theory for rotating blades in uniform flow, Proceedings of the Royal Society A:

Mathematical, Physical and Engineering Sciences 469 (2157) (2013) 20130065). However, factor 2 is the most common approach.

If the reviewer does not refer to this factor, we kindly ask for a more detailed explanation of what should be checked.

7. Line 104, "at the same relative location with respect to the observer", not a precise estimate.

What we mean by this sentence is that, assuming a wind farm, there would be a unique global observer for the wind farm. However, this observer would be different for each turbine, considering that the noise is calculated using the origin of the coordinate system in the turbine hub. Therefore, we would need to calculate the coordinates of the global farm observer for each turbine. To do so, we would need to specify a layout, which is out of the scope of this research.

We have changed the sentence to be more specific:

To compute the noise generated by a wind farm, we assume that each turbine acts as an uncorrelated noise source with equal intensity. he observer for each turbine is the same, using as the origin of the coordinate system the hub of each turbine. This results in adding a factor of  $10\log_{10} N$ , where N is the number of wind turbines.

8. Figure 2, the symbol  $\phi$  is not consistent with the one in the main text.

We apologize, we used the symbol  $\phi$  given by defect in the journal's template. For the final version of the manuscript, we will change the figure to have a symbol that is closer to the Latex version of the main text.

9. Line 153,  $d_1$  or  $d_2$ ?

We appreciate the carful revision. Indeed, there was a mistake.

We meant r from the formula right before. Later in the sentence we specify that for the air-water transmission  $r = d_1$ . We updated the text as follows:

where  $\alpha_a$  is the attenuation in dB/m r is the distance from the noise source to the observer. For the Air-side observer, r is the linear distance from the noise source to the observer, and for the Water-side observer, r is the distance from the noise source to the interface observer  $d_1$  in Figure 2).

10. Line 155, how do you consider the attenuation of sound in water?

The attenuation of sound in water is very small compared to the attenuation in air. In the water, sound attenuates between 0.1 and 1 dB/km in the range of 1 to 10 kHz. As the observer is relatively close to the wind turbine, we do not consider the attenuation of sound in water. However, this can be easily implemented. We can update the results for the final version of the paper.

11. Table 1, the numbers for IEA 22MW are different from the original definition.

We double-checked the nominal condition, and indeed, there is a mismatch between our data and the current published operational conditions. We updated the results with the nominal conditions reported in Zahle, F., Barlas, T., Lønbæk, K., Bortolotti, P., Zalkind, D.,

Wang, L., Labuschagne, C., Sethuraman, L., Barter, G., and Marten, D.: IEAWindTask37/IEA-22-280-RWT: v1.0.1, https://doi.org/10.5281/zenodo.10944127, 2024b.

12. Line 176, the definition of low-frequency sound is different from the standard definition.

We agree with the reviewer. We meant in the low-frequency range of the total range we are analysing, not the standard definition. We changed the sentence.

13. In figures, "Noise Amplitude", should it be "Overall Sound Pressure Level"?

It should be "Sound Pressure Level" instead of "Noise Amplitude". We have changed the figures. In the directivity plot (Figure 6), we used "Overall Sound Pressure Level (OSPL)" because it is integrated over the entire frequency range.

---

## Author Comment (AC2)

**Reviewer #1, comment #1**

*The paper presents a numerical model to predict the generation of noise from off-shore wind turbines due to aeroacoustic sources and the transmission of the noise into the water. The effect of this noise on the marine life is investigated. The authors conclude that aerodynamically generated noise by the wind turbines blade is a potential environmental challenge and mitigation measures such as trailing edge serrations should be applied. The authors use largely simplified models throughout the modelling chain. The uncertainty of the model is not discussed and validation is very sparse. It seems that the authors do not understand the limitations of their model. The conclusions are bold. I think a proper scientific manuscript would deserve more careful judgement and critical discussion. I recommend performing a validation of each element of the modelling chain before using it. I also recommend that the authors collaborate with marine biologists to set the predicted noise levels into the context of the underwater environment and judge the harmfulness for marine life. I strongly recommend against accepting this article for publication. Below I have listed some detailed criticism about the modelling.*

**Our response #1.1**

We appreciate the comments and questions of the reviewer. They proposed interesting discussions that aimed to increase the accuracy and precision of the model, and, consequently, the confidence on the results. We agree that our model is simple and indeed, this is one of the advantages of the methodology we propose here since it can be easily implemented in design and optimization processes. We do understand the limitations of our model and agree that our research is a first approach to model the aerodynamic noise of offshore wind turbines transmitted underwater and the potential effect in marine life. We do not expect to have the precision of high-fidelity simulations. However, we do believe that our model is valid and can be implemented.

We think that in the answers provided below, we demonstrated why each part of the modeling chain can be used for the cases we analyze in this research.

We strongly believe that collaboration among different knowledge areas would improve the rate of knowledge generation, and future collaboration with marine biologists would significantly help to understand the real harmful effects that aerodynamic noise from offshore wind turbines can cause in marine life. However, what we claim in this research is that aerodynamic noise from offshore turbines can be heard by several marine species, and this could become an environmental problem. To get to this conclusion we used high-quality published research.

**Air-water transmission model**

**Reviewer #1, comment #2**

*The model is based on the plane wave refraction theory. This is the most simple analytical model available in the literature. The plane wave approximation is known to be inaccurate at low frequencies. The analytical model for the transmission of spherical sound waves between two media [Salomons, E. M.: Computational Atmospheric Acoustics, Springer Science C Business Media, B.V., https://doi.org/10.1007/978-94-010-0660-6, 2001.] is usually more accurate than the plane wave transmission model.*

**Our response #1.2**

The reviewer is right. The plane wave theory is the simplest model, and, indeed, it can introduce some inaccuracies for very low frequencies. However, for the turbines we are analyzing, the spherical transmission loss due to the different media gives very close results to plane-wave theory. We demonstrate this statement in this answer.

As suggested by the reviewer, we can use the analytical expression for the transmission loss of the pressure for spherical and planar waves for two fluid media, calculated as:[1]

**Spherical waves:**

$$Tp = \frac{\phi_{in}\rho_2}{\phi_{out}\rho_1}; \tag{1}$$

where $\phi_{in}$ and $\phi_{out}$ are the incident and transmitted pressure waves, and $\rho_1$ and $\rho_2$, the density of the two media (in our case air and water). The incident and transmitted pressure waves are:

$$\phi_{in} = \int_0^\infty \frac{J_0(r_i x)}{q_1} e^{-|z-h|q_1} x \, dx, \tag{2}$$

$$\phi_{out} = \int_0^\infty T \cdot \frac{J_0(r_i x)}{q_2} e^{-hq_1 + zq_2} x \, dx; \tag{3}$$

where $r = h\tan(\theta)$, $h$ is the height of the noise source, $z = 0$ is the location of the interface,

and $T$ is the transmission coefficient defined as:

$$T = \frac{2\rho_1 q_2}{\rho_2 q_1 + \rho_1 q_2}; \tag{4}$$

where $q_1 = \sqrt{1 - \kappa_1^2}$ and $q_1 = \sqrt{1 - \kappa_1^2}$, and $\kappa_1$ and $\kappa_2$ are the wavenumber ($\kappa = 2\pi f/c$) of medium 1 and 2, respectively ($c$ the speed of sound of each medium). Next, we detail the transmission loss for planar waves.

**Planar waves:**

$$Tp = T\frac{\rho_2}{\rho_1}; \tag{5}$$

where $T$ is defined as:

$$T = \frac{2\rho_1 \cos\theta}{\rho_2 \cos\theta + \rho_1 \sqrt{\frac{c_1^2}{c_2^2} - \sin^2\theta}}. \tag{6}$$

The transmission loss of the sound intensity (for either implementation) can be calculated as $T_i = 10\log_{10}\left(Tp\frac{Z_1}{Z_2}\right)$, where $Z_1$ and $Z_2$ are the impedance ($Z = \rho c$) of each medium.

Figure 1 shows the transmission loss of the sound intensity for spherical and plane waves for several frequencies at two heights of the noise source, i.e., 90 m and 170 m, which are the hub heights of the shortest and tallest turbines that we are studying in our manuscript. The figure shows that the transmission loss for the spherical wave fluctuates along the angle of incidence, whereas the plane wave loss remains flat. What is important here is that the difference between the plane and spherical losses is not larger than 1.5 dB (for any incident angle). More specifically, at $\theta = 13.1\circ$, that is, the critical Snell angle for air and water, the difference is less than 0.5 dB.

To verify our previous results, we have coupled the spherical wave transmission loss with our proposed prediction method. The results are shown in Figure 2. It can be seen that there is no difference in the far-field noise spectrum when switching from plane wave theory to spherical plane for the wind turbine application at hand.

[Figure]

[Figure]

(a) distance from source to sea level: h = 90 m    (b) Distance from source to sea level: h = 170 m

Figure 1: Intensity transmission loss as a function of the incidence angle for several distances (source to sea level) and frequencies.

For completeness, Figure 3 shows the difference in the average transmission loss calculated with spherical and planar wave theory, in the entire frequency range and for each blade section at each azimuth location for both, the EIA 22MW and the NREL 5MW, the largest and smallest turbines, respectively. The difference in transmission loss is less than 0.6 dB for all the frequencies and sections.

Finally, according to the literature, spherical waves can be approximated by plane waves for long distances, i.e., ten times the wavelength (or more).[1–4] Figure 4 shows the ratio of the distance from the noise source (each blade segment at each azimuth location) and the wavelength at $f = 50$ Hz for both, the EIA 22MW and the NREL 5MW. For both turbines and all the segments, the condition of long distances (ten times the wavelength (or more)) is met, and hence considering plane waves is an accurate approximation in this context.

All in all, we believe that modeling the transmission loss for the air-water interface using plane wave theory is valid for the cases we are studying.

**Reviewer #1, comment #3**

*According to the plane wave model, total reflection occurs at an air-water interface when the angle to the surface normal is above 13 degrees. Within this narrow angular range there is a steep dependency on the angle of incidence. The authors assumed that the water can be modeled as a perfectly flat surface. This assumption needs to be verified further. Due to the steep gradients of the model sensitivity, even small surface waves in the water could significantly alter the transmission loss. Therefore, I assume that actual transmission loss would be higher than one predicted with the flat-surface model.*

**Our response #1.3**

We understand the concern of the reviewer. Assuming a flat surface is an approximation that is well-established in the text as a limitation of our work. Analyzing the effect of waves is an excellent future work; however, it is out of the scope of our work since it needs to consider additional factors, such as the frequency of the sea waves and the amplitude, among others.

Having a non-flat surface would change the angle of incidence of the sound wave, explained by Snell's law, and therefore, the noise propagated by the turbine would change due to the

[Figure]

Figure 2: IEA 22 MW wind turbine trailing-edge noise for an observer located 100 m downstream and 10 m underwater obtained with spherical and planar wave transmission loss (to account for the change in media).

[Figure]

(a) IEA 22 MW

(b) NREL 5 MW

Figure 3: Intensity transmission loss as a function of the incidence angle for several altitudes and frequencies

directvity. However, as shown in Figure 1, the angle of incidence does not significantly vary transmission loss. Therefore, the intensity would not change significantly.

Despite the fact that assuming a flat surface is an approximation, we strongly believe that this does not invalidate our work and the methodology here presented is valid for any sea state. In future work, knowing the condition of the sea where the turbines would be located, a more accurate estimate could be carried out using the methodology established in this paper, which is the main contribution of our work.

**Reviewer #1, comment #4**

*The transmission model is crucial in the modelling chain. It should be validated against measurements or high-fidelity numerical models.*

**Our response #1.4**

Our models are theoretical and we have not yet performed validations of the transmission

[Figure]

|                    |                   |
|--------------------|-------------------|
| (a) IEA 22 MW      | (b) NREL 5 MW     |

Figure 4: $r/\lambda$ for every blade section at each azimuth location at $f = 50$Hz. $r$ is the distance of the noise source to the air-water interface and $\lambda = c/f$ the wavelength.

model; however, this model has been validated with numerical approaches, such as parabolic equations, Multipath expansion, normal modes, Fourier integral and wavenumber integration .[2,3]

**Wind farm noise modeling**

**Reviewer #1, comment #5**

 The wind farm model assumes that all wind turbines are located at the same position with respect to the observer location. It is used to compute the effects of having 100 or 150 very large wind turbines in a wind farm. Such a wind farm would stretch over several kilometers (at 10x10 squared wind cluster with 4.5 rotor diameter distance between each 10 MW wind turbine would cover a space of about 8x8 km). It is unlikely that the assumption is justified for such a wind farm. At the same time the added level by this model is crucial in whether the predicted sound exceeds the hearing threshold or not.

**Our response #1.5**

 We agree with the reviewer in that the wind farm prediction can be more precise if considering particular layouts. In the new version of the manuscript, we use the wind farm layout proposed by the reviewer to predict the wind farm noise. The results are shown in Figure 5 for the EIA 22 MW. The final version of the paper will include the updated wind farm results for the three turbines.

 It is important to mention that the absolute noise level of the wind farm depends on the layout and location of the observer. However, the results of the wind farm do not affect the main conclusions of this research, as the noise level of a single turbine is higher than the noise thresholds of various marine animals.

[Figure]

Figure 5: Far-field noise of a wind farm (with EIA 22MW wind turbines) of 10x10 turbines separated by 4.5D in each direction.

**Wind turbine noise prediction code**

**Reviewer #1, comment #6**

*The prediction code has been validated by means of a single data set of noise measurements on a 2.3MW wind turbine. The differences between a 2.3MW and a 22MW wind turbine are significant. It is doubtful that one can trust the prediction model.*

**Our response #1.6**

We appreciate the reviewer's concern regarding turbine size. Our noise-prediction methodology is built on first-principles aeroacoustic theory rather than on empirical curve-fitting to a particular machine. The model computes the unsteady pressure fluctuations along the blade surfaces from the local flow and blade geometry, and then propagates those fluctuations into the far field using classic acoustic radiation integrals. All of the geometric, kinematic, and operating-condition parameters (blade chord, span, rotation rate, inflow turbulence intensity, tip Mach number, etc.) enter the model explicitly, so it is inherently scalable from small turbines to multi-megawatt machines.

Validation against the 2.3 MW dataset demonstrates that our implementation faithfully captures the underlying physics; it does not 'tune' the coefficients to that machine. Consequently, the same code, when supplied with the blade and operational parameters of a 22 MW turbine, will generate prediction spectra consistent with the new size and operating point.

**Trailing-edge noise prediction**

**Reviewer #1, comment #7**

*The Stalnov trailing edge noise model was tuned to fit measurements for low Reynolds numbers, see figure 8(a)-(d) in [Stalnov, O., Chaitanya, P., and Joseph, P. F.: Towards a non-empirical trailing edge noise prediction model, Journal of Sound and Vibration, 372, 50–68, https://doi.org/10.1016/j.jsv.2015.10.011, 2016.]. The best fit with measured data occurs at Reynolds numbers 0.26 and 0.52 million. The agreement with the measurements becomes gradually worse for Reynolds numbers 0.78 and 1.04 million and is more than 3 dB off. The blade*

*sections at the outer part of the rotor of the 22MW reference turbine operate at Reynolds number of 15-20 million. Therefore it is very unlikely that the trailing edge noise will be predicted with sufficient accuracy.*

**Our response #1.7**

The reviewer has raised a good point, which was probably not detailed in the previous version of the manuscript. There are several empirical methods for predicting trailing-edge noise and the wall pressure spectrum. However, the methods that we consider in our manuscript, i.e., Amiet's theory for predicting the noise and the TNO-Blake model for calculating the wall-pressure spectrum are semi-analytical, meaning that they are based on the theoretical considerations (and not data fitting).

We have used the version of the TNO-Blake model proposed by Stalnov et al.[5] and published in the reference mentioned by the reviewer. This method calculates the wall-pressure spectrum based on the solution of the Poisson equation. In the reference, Stalnov does not use any measurements to fit the model. Therefore, we strongly believe that this model can be used for our range of flow conditions.

Regarding the noise prediction method, i.e. Amiet's theory,[6] it is also an analytical method, and not empirical. In Amiet's theory, the Reynolds number is not as relevant as the Mach number. In subsonic flow (which is our case), the theory is valid. Amiet's theory has been used before for rotary sources with a Mach $= 0.748$ at the tip.[7] Therefore, we believe that we are still in the range where we can use Amiet's theory.

Finally, the Reynolds number does influence the bounday layer quantities required by Amiet, but these are always calculated numerically, using the XFOIL in this work. XFOIL has been verified/validated/used for a variety of high-Reynolds numbers in the literature showing accurate results.[8] Therefore, the boundary layer quantities are appropriate for the high Reynolds numbers associated with the large wind turbines considered in our study.

In summary, the methodology selected in our work to link aerodynamic quantities to acoustics is flexible in terms of flow conditions and can be used for a variety of wind turbine sizes.

**Effect on marine life**

**Reviewer #1, comment #8**

*The authors conclude that any sound that is above the threshold of hearing can be perceived. They do not assess underwater background noise levels. A more detailed study is necessary to assess the audibility of the predicted sound.*

**Our response #1.8**

In our comparison of underwater acoustic emissions from wind turbines to marine animal hearing thresholds, we deliberately omitted background noise for two main reasons.

First, ambient sound levels in the ocean fluctuate dramatically with location, water depth, seabed composition, shipping traffic, and weather conditions. Attempting to select a single background spectrum for our analysis would require assumptions so broad that they risk obscuring rather than clarifying the true impact of turbine noise. Because our work is intended to be generally applicable rather than site-specific, we avoid introducing a background noise profile that may not reflect the range of real-world environments in which the turbines operate.

Second, current offshore wind certification protocols and environmental assessments also focus on turbine source levels relative to biological hearing thresholds, without trying to account for

the ambient soundscape. This industry-standard approach ensures consistency across projects and regulatory reviews. By aligning our methodology with established certification practice, we provide results that are both directly comparable to existing benchmarks and free from the additional uncertainty that would accompany an arbitrary choice of background noise.

Finally, if background noise spectra become available to any researcher for a particular site, these can be overlapped with our results to draw site-specific conclusions.

**Statements in the conclusions**

**Reviewer #1, comment #9**

*The authors claim they have identified trailing edge noise as the dominant aerodynamic noise source to effect marine life. However, the model only considers leading edge noise and trailing edge noise. Other mechanisms are not modelled. Mechanical noise is not modelled either. Due to modelling assumptions the dominant noise sources are already determined beforehand.*

**Our response #1.9**

We thank the reviewer for highlighting the scope of our noise model. It is well established in both wind-turbine and aeroacoustics literature that, of the aerodynamic mechanisms, trailing-edge noise overwhelmingly dominates the broadband spectrum for high-Reynolds-number blades (e.g.[6,9,10] Leading-edge scattering contributes primarily at very low frequencies and under specific turbulent inflow conditions, whereas other aeroacoustic sources (e.g. laminar-turbulent transition noise, boundary-layer instabilities, and tip vortex noise) are either confined to narrow bands or occur at levels significantly below the trailing-edge component.

Mechanical noise — arising from drivetrain, gearbox and generator — is indeed an important contributor in some operational regimes, but it is typically narrow-band, low frequency and can be mitigated through standard vibration-isolation measures. Our study intentionally isolates aerodynamic noise because it sets the broadband baseline against which both biological thresholds and any mechanical contributions must be compared. Thus, by focusing on leading- and trailing-edge mechanisms, we capture the dominant aerodynamic contribution; if additional sources prove significant at a specific site, they can be added as an offset to our source-level spectra before comparison to marine-animal audiograms.

We will change the final version of the manuscript and explain better why trialing-edge noise is dominant and define the limitations of this statement.

**References**

[1] Y. Chen, Spherical wave reflection and transmission, Open University (United Kingdom), 1992.

[2] D. M. Chapman, P. D. Ward, The normal-mode theory of air-to-water sound transmission in the ocean, Journal of the Acoustical Society of America 87 (1990) 601–618. `doi:10.1121/1.398929`.

[3] D. M. Chapman, D. J. Thomson, D. D. Ellis, Modeling air-to-water sound transmission using standard numerical codes of underwater acoustics, Journal of the Acoustical Society of America 91 (1992) 1904–1910. `doi:10.1121/1.403701`.

[4] C. H. Hansen, C. J. Doolan, K. L. Hansen, Wind farm noise: measurement, assessment, and control, John Wiley & Sons (2017).

[5] O. Stalnov, P. Chaitanya, P. F. Joseph, Towards a non-empirical trailing edge noise prediction model, Journal of Sound and Vibration 372 (2016) 50–68. `doi:10.1016/j.jsv.2015.10.011`.

[6] R. Amiet, Noise due to turbulent flow past a trailing edge, Journal of Sound and Vibration 47 (3) (1976) 387–393. `doi:10.1016/0022-460X(76)90948-2`.

[7] S. Sinayoko, M. Kingan, A. Agarwal, Trailing edge noise theory for rotating blades in uniform flow, Proceedings of the Royal Society A: Mathematical, Physical and Engineering Sciences 469 (2157) (2013) 20130065.

[8] M. Drela, Xfoil: An analysis and design system for low reynolds number airfoils, Low Reynolds Number Aerodynamics 54 (1989) 1–12. `doi:10.1007/978-3-642-84010-4_1`.

[9] T. F. Brooks, D. Pope, M. A. Marcolini, Airfoil self-noise and prediction, Tech. rep., NASA Reference Publication 1218 (1989).

[10] S. Oerlemans, P. Sijtsma, B. Méndez López, Location and quantification of noise sources on a wind turbine, Journal of Sound and Vibration 299 (4) (2007) 869–883. `doi:10.1016/j.jsv.2006.07.032`.